# Altered Estrous Cyclicity and Feeding Neurocircuitry, but Not Cardiovascular Indices in Female Offspring from Dams with Previous Vertical Sleeve Gastrectomy Surgery

**DOI:** 10.3390/brainsci13081218

**Published:** 2023-08-18

**Authors:** Seth Johnson, Taylor N. Welch, Nandini Aravindan, Redin A. Spann, Bradley A. Welch, Bernadette E. Grayson

**Affiliations:** Department of Neurology, University of Mississippi Medical Center, Jackson, MS 39216, USA; sljohnson2@tougaloo.edu (S.J.); twelch5@umc.edu (T.N.W.); nandina1@umbc.edu (N.A.); redin.spann@pbrc.edu (R.A.S.); bwelch3@umc.edu (B.A.W.)

**Keywords:** diet, blood pressure, estrus cyclicity, c-Fos, appetite regulation, vertical sleeve gastrectomy

## Abstract

Metabolic syndrome (MetS), which includes obesity, diabetes, hypertension, hyperlipidemia, and fatty-liver disease, affects more than two-thirds of the U.S. population. Surgical weight loss has been popularized in the last several decades as a means to produce significant weight loss and improvements in the comorbidities of MetS. Women are by far the most common recipients of these surgeries (more than 85%). Women of childbearing age are very likely to pursue surgical weight loss to improve their reproductive function and fertility for childbearing purposes. Significant research using pre-clinical models from our laboratory and clinical data from around the world suggest that surgical weight loss before pregnancy may have negative consequences for offspring. The present study investigates the metabolic endpoints in female-rodent offspring born to dams who had previously received vertical sleeve gastrectomy (VSG) before pregnancy. Comparisons were made to offspring from lean and obese dams. In the adult offspring of either maternal VSG or sham surgery, no differences in body weight, body fat, or lean body mass between groups were identified. The blood pressure measured in a subset of female offspring showed no differences between the VSG and the sham groups. Estrus cyclicity measured by lavage on serial days showed altered cycles in the VSG offspring compared to the controls. For animals that had previously only been exposed to chow, rats were fasted overnight and then given a 1 g meal of either chow or a novel high-fat diet (HFD). The animals were euthanized and paraformaldehyde (PFA)-perfused to perform brain immunohistochemistry for c-Fos, an immediate–early gene activated by novel stimuli. In the VSG rats exposed to either the chow or the HFD meal, the c-Fos-activated cells were significantly blunted in the nucleus of the solitary tract (*p* < 0.05), the paraventricular nucleus of the hypothalamus (PVN) (*p* < 0.05), and the dorsal medial nucleus of the hypothalamus (DMH) (*p* < 0.05) in comparison to the sham controls. These data suggest that the hypothalamic wiring within the brain that controls the response to nutrients and reproductive function was significantly altered in the VSG offspring compared to the offspring of the dams that did not receive weight-loss surgery.

## 1. Introduction

Currently, over 275,000 individuals per year obtain surgical weight-loss procedures, of whom nearly 90% are female, half of whom are of child-bearing age [1]. An added incentive for obtaining bariatric surgery during the child-bearing years is the hope of improving reproductive parameters, and, hence, achieving a healthier pregnancy. According to the American Society for Metabolic and Bariatric Surgery, individuals with comorbidities other than Type-2 Diabetes Mellitus (T2DM) must have a body-mass index (BMI) greater than 35 kg/m^2^. If an individual does have T2DM, then bariatric surgery is recommended if they have a BMI greater than 30 kg/m^2^. When an individual undergoes bariatric surgery, specifically vertical sleeve gastrectomy (VSG), approximately 80–90% of the stomach is removed along its greater curvature. This process limits the intake of both liquid and solid nutrients that may be ingested at any given interval. During this procedure, the remainder of the gastrointestinal tract is undisturbed. Substantial improvements in both metabolic and reproductive health are realized through obtaining bariatric surgery [2,3,4,5]. The improvements to reproductive parameters after bariatric surgery that have been reported to date include the return of normal menstrual cycles [6], improved levels of reproductive hormones [7], the recovery of luteal function [8], increased spontaneous (unassisted) pregnancies [9], a lower risk of gestational diabetes and preeclampsia [10,11], and a reduced risk of large-for-gestational age (LGA) babies [10,12,13,14]. Currently, 70.6 per 100,000 individuals receive bariatric surgery. These rates have remained steady throughout the past 10 years. Overall, these positive outcomes continue to drive the desire to obtain surgical weight-loss procedures to enhance reproductive capability and improve the comorbidities associated with obesity.

Pregnancies preceded by surgical weight-loss procedures promise a reduction in LGA births. Unfortunately, an increased risk of small-for-gestational age (SGA) births is reported. In fact, there is a 100-percent increase in SGA births following bariatric surgery when compared to obese controls [15,16]. This was further confirmed by a meta-analysis, which agreed with the increased risk of complications, such as SGA, following surgical weight loss [17]. Because low birth weight predicts metabolic health problems such as obesity, T2DM, dyslipidemia, and cardiovascular disease later in life [18,19,20], it is imperative that the mechanisms linking birth weight and metabolic disease are investigated with respect to bariatric surgery.

Some of the complications of maternal bariatric surgery followed by pregnancy have been examined using the rodent vertical-sleeve gastrectomy (VSG) procedure followed by pregnancy. The investigation of post-VSG dams demonstrated an array of metabolic and hormonal changes in these affected females [21,22]. The gestational outcomes in VSG pregnancies in rats have significant challenges that include poor in utero health [22] and reduced postnatal growth [23]. We have described macronutrient and hormone differences in the milk produced by dams with previous VSG surgery compared to controls [24]. Both ghrelin [25] and leptin [26] signaling are dysregulated in the offspring of dams with bariatric surgery prior to pregnancy. Further, VSG offspring exhibit profound changes to their immune systems as a result of being born to dams that have lost weight through bariatric means [27]. This is not surprising, given the profound impact of VSG on the lymphoid organs and immune system following VSG [22,28]. Further, as in the current paper, we previously reported metabolic and immune problems in adult offspring [22,23,24,25,26,27]. This model has provided a durable method to investigate long-term metabolic dysfunction in VSG offspring following maternal surgical weight loss. 

In female rats that have previously received VSG surgery, cardiovascular [23], reproductive [23], and metabolic [23] indices have been reported. In the present study, the focus is on the female offspring of the female recipients of VSG or sham controls. Female offspring are born with SGA [23] and are chronically smaller during postnatal growth [23]. When challenged with an obesogenic diet for 8–10 weeks after weaning, they grow to be more obese than obese controls [23]. In the present study, the females were weaned onto a chow diet to limit the diet-induced effects we previously caused to appear. The female offspring were investigated under standard chow conditions only to determine the effects without the further manipulation of their baseline metabolism. The rats’ blood pressure was monitored and the plasma analytes were measured. The hematoxylin and eosin (H&E)-stained vaginal cells were rated for estrus-cycle regularity over a period of eight days. We further examined the appetitive system by interrogating the rats’ responsivity to a small meal, as reflected by the 90-minute c-Fos immunoreactivity following ingestion. Finally, relevant hypothalamic gene expression was investigated for targets controlling food intake and body weight. Overall, this study supports the hypothesis that bariatric surgery prior to pregnancy results in long-term changes to female offspring. This means that mothers’ decision to improve health by obtaining bariatric surgery does not solely affect the trajectory of their own health, but also alters the reproductive cyclicity and food-intake neurocircuitry in offspring born after the VSG procedure. This work demonstrates the transgenerational effect of maternal surgical weight loss on parameters that are likely to affect the fecundity and health of female progeny.

## 2. Materials and Methods

### 2.1. Animal Assurance

All animal-use protocols were in accordance with the National Research Council of the National Academies’ Guidelines for the Care and Use of Laboratory Animals. All methods were evaluated and approved by the University of Mississippi Medical Center Institutional Animal Care and Use Committee, Protocol #2014-1423.

### 2.2. Animals

Generation of dams to produce offspring for this study. Females (200–250 g) and males (300–350 g) of the Long–Evans strain (RRID:RGD5508398, Envigo, Indianapolis, IN, USA) were kept in standard plexiglass cages in a room with a 12 h light/12 h dark cycle, 23.0 2 °C temperature, and 50–60% humidity, with ad libitum access to water and standard rat chow (Teklad #8640, Madison). Animals were originally housed in groups and then separately after operation. 

All female rats were maintained on a palatable high-fat diet (#D03082706, Research Diets, New Brunswick, NJ, USA, 4.54 kcal/g; 40% fat, 46% carbohydrate, 15% protein) for four weeks before surgery after one week of acclimation to the environment. The rats were subsequently separated into two weight-matched groups: sham, N = 20, and VSG, N = 24. These bariatric animals were previously well characterized [26,27] and were used to generate the female offspring who were subjects used in this study.

#### 2.2.1. Surgical Procedures

Animal care was performed as previously described [22,25,26,27]. Briefly, animals were maintained on Osmolite OneCal liquid diet (Abbott Laboratories, North Chicago, IL, USA). In order to encourage a clear digestive tract, no-solid diet was provided 24 h before surgery. For VSG surgery, rats were anesthetized with isoflurane (Piramal Enterprises, Ltd., Andhra Pradesh, Telegana, India). Vertical sleeve gastrectomy involved a medial abdominal incision through the skin and muscular layer. The ligaments were cut to produce stomach externalization. To remove the lateral 80% of the stomach, an ENDO GIA Ultra Universal stapler (#EGIAUSHORT, Covidien, Mansfield, MA, USA) with an ENDO GIA Auto Suture Universal Articulating Loading Unit, 45 mm–2.5 mm (#030454, Covidien, Waltham, MA, USA) was used to reconfigure the now tubular stomach. The gastric sleeve was replaced in the abdominal cavity and sutured with 4-0 vicryl suture. For sham VSG, following laparotomy, the stomach was externalized and forcefully pressed between blunt forceps for 15 s, and then reintegrated into the abdominal cavity.

#### 2.2.2. Post-Operative Care 

Animals were maintained on Osmolite OneCal liquid diet for three days and then transitioned to chow diet for the remainder of the study. Animals received 5 ml of 0.9% saline daily with buprenorphine and carprofen for pain management for three days. Animals were weight-stabilized for six weeks before mating, as previously described [22,25,26,27].

Breeding Protocol to Generate Female Offspring for this Study. Male Long–Evans rats were purchased for study purposes from Envigo (N = 20) and housed singly for harem mating (2–3 sham or VSG females, described above, per male). Female VSG or sham rats were housed singly once pregnancy was ascertained and allowed to give birth naturally. The dams reared their pups, who were weaned on postnatal day 21 (PND21) and maintained on either chow or HFD diet until PD75. For these studies, N = 17 chow–Sham, N = 15 chow–VSG, N = 20 HFD–sham, and N = 15 HFD–VSG female offspring were used. Female offspring were studied on PND 75–80.

#### 2.2.3. Blood Pressure (BP) Determination 

A subset of rats, approximately PND 75, were anesthetized by isoflurane-gas anesthesia, and a catheter was surgically placed in the femoral artery for blood sampling and BP monitoring. The catheter was exteriorized at the back of the neck. The next day, conscious BP recordings were taken from rats placed in restraining cages. Rats were habituated to the restraining cages before catheter placement. Mean arterial BP was monitored in conscious rats with a pressure transducer connected to a Grass recorder (model 7B-chart, Grass Instrument). After a 60 min stabilization period, recordings were made for two periods of 30 min each, and the data were averaged.

#### 2.2.4. Vaginal Lavage and Determination of Estrus Cyclicity

In a subset of animals, daily estrus-cyclicity checks were performed through vaginal lavage. Warmed saline was gently pipetted into the vaginal canal two hours after lights-on. Vaginal cells were collected on glass slides from rat lavages and allowed to dry. The cells were dipped in PFA, and then stained in the standard manner with H&E. The stained cells were imaged with a microscope and matched to the standardized images of the four phases of the estrus cycle.

#### 2.2.5. Novel Diet-Induction Test

In a subset of chow-fed-only rats, animals were fasted overnight and transported to the testing room 1.5 h after lights-on. A test meal of either standard rat chow or novel HFD pellet (1 g) (Research Diets, New Brunswick, NJ, USA, 4.54 kcal/g; 40% fat, 46% carbohydrate, 15% protein) was administered in a staggered fashion. Rats remained in their cages for 90 min until they were injected with Fatal-Plus and perfused with a saline flush followed by 4% PFA. Brains were extracted and placed in PFA for 48 h and then transferred to 30% sucrose in water until sectioning.

#### 2.2.6. Immunohistochemistry

To determine which nuclei in the hypothalamus were responsive to test-pellet consumption, we used c-Fos as a marker of neuronal excitation, as previously performed in a laboratory [26]. Brains were flash-frozen in methylbutane on dry ice and then sectioned at 30 microns using a freezing–sliding microtome. Sections were washed in potassium-phosphate-buffered saline (KPBS), blocked with 2% donkey serum in KPBS with 0.4% Triton x-100 in KPBS, and incubated with the primary antibody for c-Fos (1:2500, #ab190289, Abcam, Cambridge, MA, USA) overnight for two nights at 4 °C. The sections were then incubated with the anti-rabbit biotinylated secondary antibody (1:500, #BA1100, Vector Laboratories, Burlingame, CA, USA) for 60 min at room temperature, followed by A/B solution (1:600, #PK-6100, Vector Laboratories) for 60 min at room temperature. The final step was an incubation of 0.2 mg/mL DAB substrate (D5905, Sigma, Burbank, CA, USA) and hydrogen peroxide for 15 min at room temperature. Sections were then mounted on gelatin-coated slides.

#### 2.2.7. Microscopy and ImageJ Cell Counting 

Images were taken by an investigator (blinded) using an Olympus BX60 F5 light microscope with a Leica DFC310 FX camera and Leica Application Suite software version 4.6 (Leica Microsystems, Buffalo Grove, IL, USA). Brain regions were identified using Paxinos et al., *The Rat Brain*, using sections that were matched between Bregma −1.80 and −3.48 mm (arcuate nucleus (ARC), dorsomedial nucleus of the hypothalamus (DMN), ventromedial nucleus of the hypothalamus (VMH)), and between Bregma −13.92 mm and −14.28 mm for the nucleus of the solitary tract (NTS). Once the photomicrographs were obtained, the images were processed using ImageJ. The counting function in ImageJ was used to count the cells. The c-Fos-positive cells were independently counted for the right and the left and then averaged with counts from 2–3 other sections in the same region.

#### 2.2.8. Fresh Tissue Harvesting 

Animals were euthanized by conscious decapitation. Trunk blood was collected in EDTA tubes and spun at 5000 RPM in a clinical centrifuge. Plasma was pipetted to new tubes and stored at −80 °C until assayed. Brains were extracted and flash-frozen in methylbutane on dry ice and stored at −80 °C until further processing.

#### 2.2.9. Triglyceride Assay 

Triglycerides were measured according to the manufacturer’s specifications (#TR22421, ThermoFisher, Richmodn, VA, USA). Blood was diluted in saline, 1:20 and a standard curve produced using known standards. The colorimetric assay was read on a spectrophotometer at 540 nm and OD readings used to extrapolate plasma triglyceride concentrations. 

#### 2.2.10. RNA Processing and Real-Time PCR

Brains were microdissected, and the medial–basal hypothalamus was prepared for RNA isolation. Total RNA was isolated with TRIzol^®^ and withdrawn using a QIAGEN miniprep RNA kit (#74104, QIAGEN, Inc., Valencia, CA, USA). The cDNA was transcribed using an iScript cDNA synthesis kit (#1708891, Bio-Rad Laboratories, Hercules, CA, USA), with 2 µg of RNA per reaction. Quantitative real-time polymerase chain reaction was performed with a BioRad CFX96 Touch Real-Time PCR Detection System using TaqMan gene-expression assays (Life Technologies, Foster City, CA, USA). The final cocktail used for RT-PCR included 2 µL of cDNA (50 ng/µL), 0.5 µL probe, 5 µL Master Mix and 2.5 µL DNAase-RNAse-free water for a 10 µL reaction.

#### 2.2.11. Statistics

All statistics were derived using GraphPad Prism version 9.1.2 (GraphPad Software, San Diego, CA, USA). Statistical significance was determined with a Student’s T test, or two-way analysis of variance followed by Tukey’s post hoc test for variables of maternal surgery and diet. To determine the percentage of rats with irregular cycling, a chi-squared test was used. All results are given as means ± standard error of mean (SEM). Results were considered statistically significant when the value of *p* is < 0.05.

## 3. Results

The animals were maintained on chow for approximately 70 days. The terminal body weights were similar among the four groups (Table 1).

No differences between the groups were measured for body fat, fat-mass percentage, lean mass, or lean-mass percentage (Table 1). The measurement of the average arterial blood pressure was elevated for the HFD-fed females compared to the chow-fed controls, *p*(diet) < 0.05, A vs. B (Figure 1A).

During the active (wake) phase, the arterial pressure was also elevated in the HFD-fed animals compared to the controls, with *p*(diet) < 0.05, A vs. B (Figure 1B). Similarly, during the inactive (sleep) phase, the HFD-fed animals had higher blood pressure than the controls, with *p*(diet) < 0.05, A vs. B (Figure 1C). In addition, the terminal heart weight normalized to body weight trended to be lower in the HFD-fed animals compared to the controls, with *p*(diet) = 0.056 (Figure 1D).

Daily lavages were obtained, and the cytology was examined to categorize each female’s day in the estrus cycle. Representative images of each day of estrus are presented (Figure 2A).

The HFD-fed females exhibited high irregularity compared to the chow-fed controls (Figure 2B). The offspring of the VSG dams spent a higher percentage of time in proestrus, with *p*(surgery) < 0.01, A vs. B and * *p* < 0.05, chow–sham vs. chow–VSG (Figure 2C). In parallel, the female VSG offspring spent reduced time in the estrus phase, with *p*(surgery) < 0.01, A vs. B and * *p* < 0.05, HFD–sham vs. HFD–VSG (Figure 2D). Overall, the HFD-fed rats spent less time in metestrus, with *p*(diet) < 0.05, A vs. B (Figure 2E). Finally, no differences were observed in terms of the time spent in the diestrus phase (Figure 2F). 

The fasted animals were given a one-gram test meal of either chow or HFD, and 90 min later, the animals were euthanized to obtain PFA-perfused brains for c-Fos. regardless of whether they were fed chow or HFD, the VSG animals exhibited significantly less c-Fos in the DMH compared to the sham controls, with *p*(surgery) < 0.05, A vs. B and * *p* < 0.05, chow–sham vs. chow–VSG (Figure 3A–E) and VMH, *p*(surgery) < 0.01, A vs. B and * *p* < 0.05, HFD–Sham vs. HFD–VSG (Figure 3F–J). No differences in c-Fos levels were observed in the ARH (Figure 3L–O).

A similar pattern of reduced c-Fos levels was also observed in the NTS, with the VSG offspring demonstrating lower levels in comparison to the sham offspring *p*(surgery) < 0.05, A vs. B and * *p *< 0.01, chow–sham vs. chow–VSG (Figure 4A–F).

Additional animals were generated to determine the gene expressions of relevant proteins and their receptors in a macro-dissection of the medial–basal hypothalamus (MBH) (Table 2).

The related peptides, AGRP and NPY, were significantly elevated in the VSG offspring compared to the sham offspring, with *p*(surgery) < 0.01 and *p*(surgery) < 0.05, respectively (Table 2). The neuropeptide POMC showed no differences between the groups (Table 2). To determine whether the differences in glucocorticoid receptor (GR) expression contributed to the reported outcomes, the GRs were measured in the MBH. The GR was reduced in the HFD-fed offspring in comparison to the chow-fed offspring, *p*(diet) < 0.05 (Table 2). Next, the influence of the glucose transporters and fat-metabolism enzymes on the feeding-behavior outcomes were investigated. There were no significant differences in GLUT 1,2 and 4 expression; however, the insulin-growth-factor 2 receptor and very long Acyl-CoA synthetase were significantly reduced in the high-fat-fed rats in comparison to the controls, with *p*(diet) < 0.05 (Table 2). The marker of inflammation, interleukin-6, was significantly elevated in the VSG offspring in comparison to the sham controls, with *p*(surgery) < 0.01 (Table 2). 

## 4. Discussion

Bariatric surgery for the significant loss of excess body weight has gained in popularity over the last decades. For women of childbearing age, the potential to improve reproductive outcomes and obtain an unassisted pregnancy is attractive, in parallel with significant improvements to maternal health indices. An ongoing clinical question is the long-term health of offspring born after maternal bariatric surgery. Large cohort studies with longitudinal follow-ups are currently in progress. Thus, the majority of the outcome data are derived from animal studies. In the present work, the diet consumed by the offspring drove the cardiovascular outcomes, and few changes were produced by maternal surgery. Still, maternal surgical weight loss may significantly alter the neural networks that constitute appetite regulation. This is also true for reproductive indices. Overall, maternal bariatric surgery may substantially change the output of hypothalamic function (Figure 5, Summary Diagram).

A large body of work exists on in utero conditions and long-term cardiovascular outcomes. Rooted in the Dutch Famine studies, investigations have revealed that first-trimester malnutrition results in significant cardiovascular, glycemic control, and obesity problems later on for the offspring [29]. Furthermore, over-nutrition during gestation can result in long-term metabolic-control challenges for the offspring [30]. In prior work, the blood pressure in pregnant and nonpregnant rats with previous VSG and compared to lean and obese controls was investigated. The HFD consumption by the females was the strongest indicator of elevation in blood pressure before or during pregnancy [22]. 

Furthermore, VSG offspring were born significantly smaller than counterparts from lean or obese dams [22]. Here, an initial characterization of the adult female offspring of maternal VSG was made; there was little-to-no pre-gestational weight-loss effect on the offspring’s blood pressure. However, the offspring’s diet during the adult growth period predicted the effects on blood pressure most directly. Thus, lifestyle factors (diet quality and exercise) remain modifiable risk factors for the determination of long-term health outcomes in these offspring.

With increasing obesity, reproductive fecundity diminishes. This relationship drives women of childbearing age to obtain surgical weight-loss procedures. Surgical weight loss is highly likely to normalize the cyclicity of reproductive hormones effectively and improve successful conception in both males and females [31,32] however, there is still a need for robust data sets supporting the extent and longevity of reproductive improvements. In previous studies conducted at this laboratory, investigating cyclicity in virgin female rats after obtaining VSG, obese controls had higher rates of abnormal estrus cyclicity. Meanwhile, lean and VSG females maintained on either chow or HFD had improved (normal) estrus cyclicity. The offspring of the VSG dams experienced a uniquely differing effect on estrus cyclicity. The current work suggests that maternal VSG alters the normal estrus cycle, producing a dysregulated pattern regardless of whether the female offspring consume chow of HFD during their adult life. These data suggest that in utero exposure to fluctuations in metabolically active hormones [23,25,26], variations in nutrient availability [22,33], and immune perturbations [27] may chronically alter reproduction in VSG offspring. 

Previous studies investigated leptin and ghrelin systems in offspring following maternal VSG. Male VSG offspring exhibited a dysregulated response to exogenous leptin, failing to respond with canonical anorexia after a single dose [26]. In addition, the reaction of the PVN and VMN in the VSG offspring was heightened neuronal activation in comparison to controls [26]. Thus, the physiological response to leptin was uncoupled in the VSG offspring. Similarly, VSG male pups had reduced responsivity to exogenous ghrelin, failing to exhibit the hyperphagia of the sham controls, both during postnatal life and in adulthood [25]. The females lacked this responsivity difference, thus highlighting the sexual dimorphism of the ghrelin system. In the current work, the focus is on female offspring, and blunted responses to the test meals, which were either contextually novel (HFD) or routine (chow), were observed. The females increased in body weight to equal extents throughout their growth trajectories. This altered response to the diet implies reduced control of normal body-weight regulation. However, the impact of consuming an apportioned meal on hypothalamic and NTS neural circuitry suggests alterations in the wiring of the appetite networks. Over time, VSG females may acquire the habit of over-feeding simply because the feedback controls within the hypothalamus are dysregulated. From the paradigm set in the current studies, it is not possible to determine whether the difference in response is because of the altered firing of the vagal afferents within the gastric mucosa to the test meal or the dampened response of nutrient receptors within the gut.

## 5. Caveats and Future Directions

A deeper understanding of the longitudinal changes in offspring born to maternal VSG is required to establish whether the positives of maternal weight loss outweigh the adverse outcomes in terms of the transgenerational transmission of metabolic effects to offspring. No differences in cardiovascular measures were observed. The use of more sensitive radiotelemetry devices may have provided greater precision in the analysis of the animals’ blood-pressure fluctuations. However, the overall finding that there were no differences suggests that there may be few biologically significant variations. Concerning reproductive fecundity, the most robust test of fertility would be the measurement of time to conception. This type of study requires large cohorts of animals, and it was not feasible within the current scope of our work. The present work did not track the altered responsivity to the test diet during postnatal life, but focused exclusively on an adult time point. We did not perform parallel studies on the male offspring. However, the previous work suggests that males, and not females, are uniquely differentially responsive to important hormones that govern body weight, like leptin and ghrelin. Because the popularity of surgical weight-loss procedures has multiplied in the last decade, the effects of the surgery on vulnerable populations, such as the offspring of those obtaining the surgeries, are of great long-term value.

## Figures and Tables

**Figure 1 brainsci-13-01218-f001:**
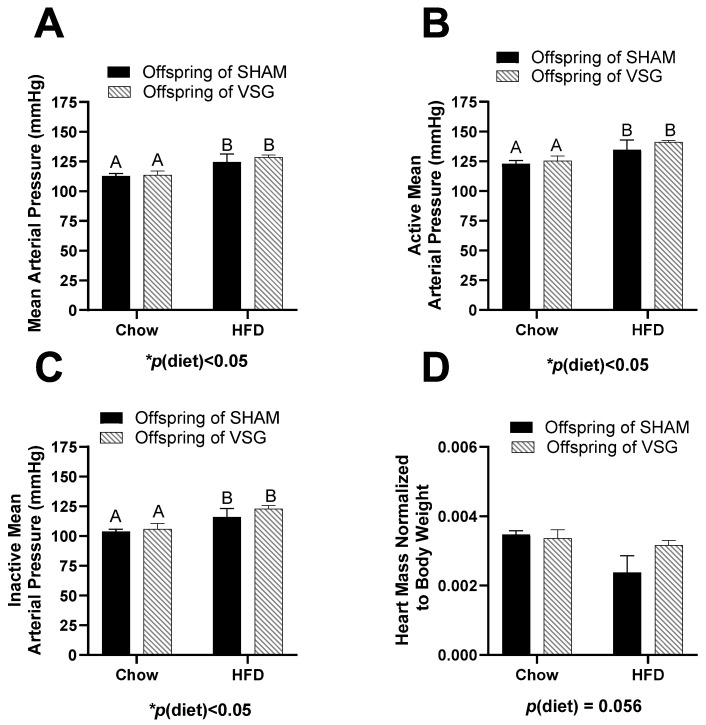
Cardiovascular measures in female offspring of maternal VSG. (**A**). Mean arterial blood pressure in mmHg, main effect of maternal diet, A vs. B, * *p* < 0.05. (**B**). Mean arterial blood pressure during the active phase, main effect of diet, A vs. B, * *p* < 0.05. (**C**). Mean arterial blood pressure during the inactive phase, main effect of diet, A vs. B * *p* < 0.05. (**D**). Terminal heart weight normalized to body weight. Data are presented as mean ± SEM and analyzed by two-way ANOVA. Main effects are reported.

**Figure 2 brainsci-13-01218-f002:**
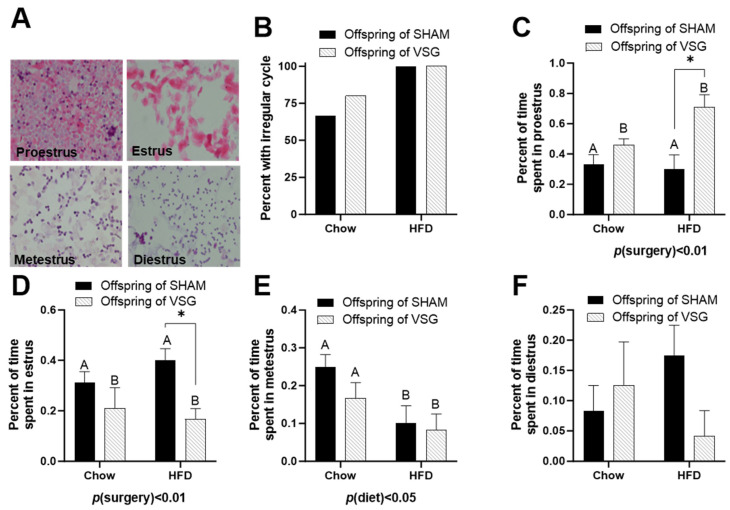
Estrus-cycle analysis in female offspring of maternal VSG. (**A**) Representative images of vaginal lavage counterstained with hematoxylin and eosin for diestrus, proestrus, estrus, and metestrus. (**B**) Percentages of animals with irregular cyclicity. (**C**) Percentages of time spent in proestrus, main effect of maternal surgery (A vs. B), * *p* < 0.05. (**D**) Percentages of time spent in estrus. * *p* < 0.05. (**E**) Percentages of time spent in metestrus, main effect of maternal diet (A vs. B). (**F**) Percentages of time spent in diestrus. Data are presented as mean ± SEM and analyzed by two-way ANOVA. A: Offspring of SHAM, B: Offspring of VSG.

**Figure 3 brainsci-13-01218-f003:**
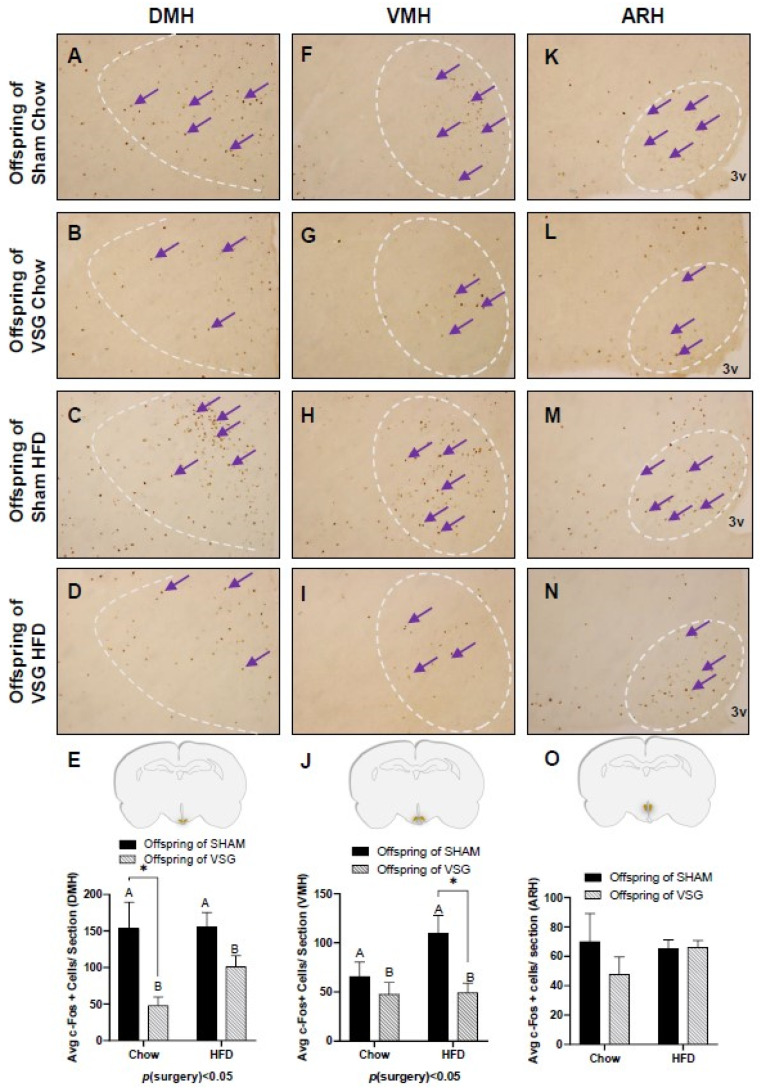
The c-Fos immunohistochemistry in the medial–basal hypothalamus (MBH) following one-gram-meal exposure. Representative images of c-Fos-positive nuclei in the dorsal medial nucleus of the hypothalamus (DMH), ventral medial nucleus of the hypothalamus (VMH), and arcuate nucleus of the hypothalamus (ARH) for sham–chow (**A**,**F**,**K**), VSG–chow (**B**,**G**,**L**), sham–HFD (**C**,**H**,**M**), and VSG–HFD (**D**,**I**,**N**). (**E**) Average c-Fos quantification for DMH, main effect of maternal surgery (A vs. B), * *p* < 0.05, individual effect; Student’s *t* test, * *p* < 0.05. (**J**) Average c-Fos quantification for VMH, main effect of maternal surgery (A vs. B), * *p* < 0.05, individual effect; Student’s *t* test, * *p* < 0.05. (**O**) Average c-Fos quantification for ARH. Data are presented as mean ± SEM. Arrows point to a selection of the many immunoreactive c-Fos-positive cells. A: Offspring of SHAM, B: Offspring of VSG.

**Figure 4 brainsci-13-01218-f004:**
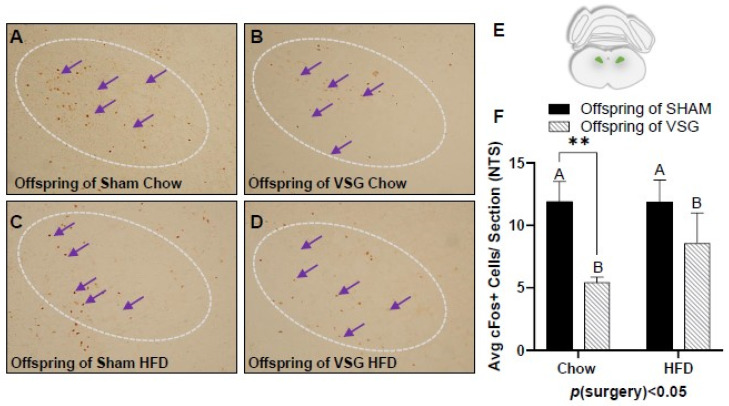
The c-Fos immunohistochemistry in the nucleus of the solitary tract (NTS) following one-gram-meal exposure. Representative images of c-Fos-positive nuclei in the NTS for offspring of sham–chow (**A**), VSG–chow (**B**), sham–HFD (**C**), and VSG–HFD (**D**). (**E**) Diagram of the sampling location for the NTS. (**F**) Average c-Fos quantification for NTS, main effect of maternal surgery (A vs. B), ** *p* < 0.01 Data are presented as mean ± SEM and analyzed by two-way ANOVA. Main effects of maternal surgery are reported (A vs. B). Arrows point to a selection of the many immunoreactive c-Fos-positive cells. A: Offspring of SHAM, B: Offspring of VSG.

**Figure 5 brainsci-13-01218-f005:**
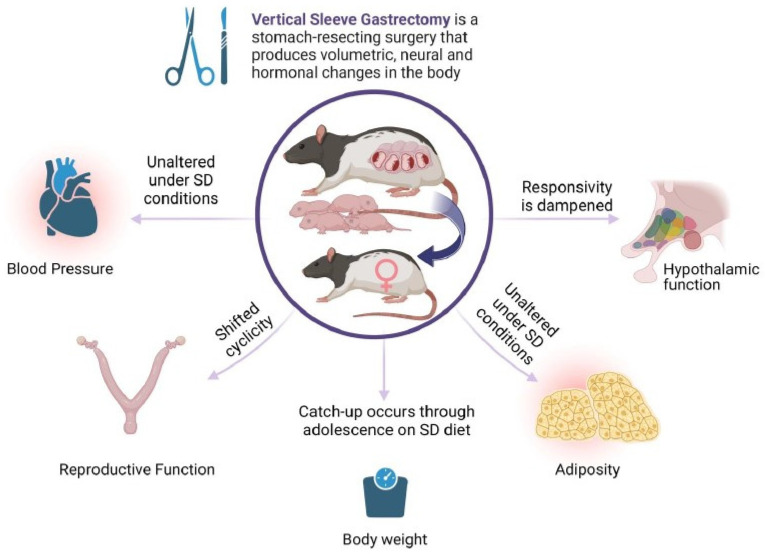
Maternal VSG produces diverse effects on core body systems in first-generation female offspring. Female offspring of females with previous VSG surgery have minimal effects on blood pressure and adiposity, but have altered reproductive function and dampened responses to meal ingestion. Image produced by BioRender^©^.

**Table 1 brainsci-13-01218-t001:** Body Weight and Composition Parameters. Data are presented as Mean ± SEM and analyzed by two-way ANOVA.

Measure	Offspring of Sham-Chow			Offspring of VSG-Chow			Offspring of Sham-HFD			Offspring of VSG-HFD			Statistics
	Mean	±	SEM	Mean	±	SEM	Mean	±	SEM	Mean	±	SEM	2W ANOVA
Body Weight (g)	335.60	*±*	5.74	314.70	*±*	10.32	328.60	*±*	11.31	318.90	*±*	30.11	*NS*
Body Fat (g)	81.36	*±*	6.18	75.76	*±*	11.36	80.06	*±*	10.22	87.58	*±*	22.31	*NS*
Fat Mass Percent (%)	24.04	*±*	1.52	23.55	*±*	2.99	23.91	*±*	2.59	26.41	*±*	4.75	*NS*
Lean Mass (g)	176.70	*±*	2.51	168.50	*±*	5.47	169.50	*±*	4.78	160.20	*±*	13.69	*NS*
Lean Mass Percent (%)	52.86	*±*	1.51	54.02	*±*	2.69	51.82	*±*	1.76	50.42	*±*	1.03	*NS*

**Table 2 brainsci-13-01218-t002:** Gene expression for the medial–basal hypothalamus by rt PCR. Data are presented as mean SEM and analyzed by two-way ANOVA.

Gene Name	Sham-Chow			VSG-Chow			Sham-HFD			VSG-HFD			Statistics
	Mean	±	SEM	Mean	±	SEM	Mean	±	SEM	Mean	±	SEM	2W-ANOVA
**Agouti-related peptide (AGRP)**	1.00	*±*	0.15	0.99	*±*	0.12	0.48	*±*	0.18	1.73	*±*	0.42	** *p*(surgery) < 0.01
**Glucocorticoid Receptor (NR3C1)**	1.00	*±*	0.16	1.16	*±*	0.10	1.13	*±*	0.27	0.72	*±*	0.21	* *p*(diet) < 0.05
**Glucose Transporter 1 (GLUT-1)**	1.00	*±*	0.16	1.07	*±*	0.06	0.97	*±*	0.30	0.51	*±*	0.16	NS
**Glucose Transporter 2 (GLUT-2)**	1.00	*±*	0.16	1.16	*±*	0.10	1.13	*±*	0.27	0.72	*±*	0.21	NS
**Glucose Transporter 4 (GLUT-4)**	1.00	*±*	0.09	0.97	*±*	0.07	0.93	*±*	0.14	0.75	*±*	0.19	NS
**Insulin Growth Factor 2 Receptor (IGF2R)**	1.00	*±*	0.14	1.01	*±*	0.06	0.88	*±*	0.14	0.50	*±*	0.14	* *p*(diet) < 0.05
**Interleukin 6 (IL-6)**	1.00	*±*	0.08	1.17	*±*	0.13	0.62	*±*	0.17	1.30	*±*	0.14	** *p*(surgery) < 0.01
**Neuropeptide Y (NPY)**	1.00	*±*	0.15	0.96	*±*	0.11	0.42	*±*	0.13	1.31	*±*	0.26	* *p*(surgery) < 0.05
**Pro-opiomelanacortin (POMC)**	1.00	*±*	0.23	0.98	*±*	0.22	0.31	*±*	0.16	0.92	*±*	0.23	NS
**Very long-chain acyl-CoA synthetase (SLC27A)**	1.00	*±*	0.15	1.07	*±*	0.07	0.88	*±*	0.23	0.49	*±*	0.14	* *p*(diet) < 0.05

## Data Availability

Data sharing is available upon direct written request.

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
