# Peer review of "Altered Estrous Cyclicity and Feeding Neurocircuitry, but Not Cardiovascular Indices in Female Offspring from Dams with Previous Vertical Sleeve Gastrectomy Surgery"

_brainsci, 2023, doi:10.3390/brainsci13081218_

Round 1

Reviewer 1 Report

The title is not clear. It does not mention the experimental model; human, mouse, or something else, or several. Please clearly mention so that it would be easier to the readers and not misleading.

Please show vertical sleeve gastrectomy, hypothalamic wiring within the brain in figure or animated figure for better understanding.

Line 44, BMI greater than 30 kg/m2When. Please put a (.) before When.

Line 55, Today, 70.6 per 100,000 individuals receive bariatric surgery today. Please remove one today from the sentence as there are 2.

Line 62, when compared to obese controls. (15, 16)... The full stop should be after the citations.

Line 108-110, To remove the lateral 80% of the stomach, an ENDO GIA Ultra Universal stapler (#EGIAUSHORT, Covidien, MA) with an ENDO GIA Auto Suture Universal Articulating Loading Unit, 45 mm – 2.5 mm (#030454, Covidien, MA). I don't understand the sentence. Please explain and complete the sentence.

Please provide figure (animated or real) for the Surgical Procedures, Blood Pressure (BP) Determination, Microscopy and ImageJ Cell Counting. 

Male Long Evans rats were purchased from Envigo mentioned in the section Breeding Protocol. I think it should be mentioned earlier when the rats were mentioned for the first time.

 Line 156, blocked with 2% donkey serum in KPBS with triton x-100. Please mention the percentage of triton x-100 and how long they were blocked.

Please mention the amount of RNA used in RNA processing and real-time PCR.

Line 213-216, Figure 1. CardiovascularMeasures.A)MeanarterialbloodpressureinmmHg.(B)Meanarterialbloodpressureduringtheactivephase.C)Meanarterialbloodpressureduringtheinactivephase.D)Terminalheartweightnormalizedtobodyweight.*p<0.05.Dataarepresentedasmean±SEMandanalyzedbytwowayANOVA.Maineffectsofdietsandmaternalsurgeryarereported. Please....

Please clearly show the significance (i.e., *) on all figures, clearly show compared to which group the data is significant.

Immunohistochemistry figures are not clear enough, please provide big clear images, label them for the staining beside the images, mention or show the changes in the figure by using arrows or some other things.

Please check the text and make corrections.

Author Response

The title is not clear. It does not mention the experimental model; human, mouse, or something else, or several. Please clearly mention so that it would be easier to the readers and not misleading.

This has been changed to reflect the work

Please show vertical sleeve gastrectomy, hypothalamic wiring within the brain in figure or animated figure for better understanding.

We have added a formal summary diagram

Line 44, BMI greater than 30 kg/m2When. Please put a (.) before When.

This has been changed.

Line 55, Today, 70.6 per 100,000 individuals receive bariatric surgery today. Please remove one today from the sentence as there are 2.

This was deleted

Line 62, when compared to obese controls. (15, 16)... The full stop should be after the citations.

This period was deleted

Line 108-110, To remove the lateral 80% of the stomach, an ENDO GIA Ultra Universal stapler (#EGIAUSHORT, Covidien, MA) with an ENDO GIA Auto Suture Universal Articulating Loading Unit, 45 mm – 2.5 mm (#030454, Covidien, MA). I don't understand the sentence. Please explain and complete the sentence.

This sentence was completed appropriately

Please provide figure (animated or real) for the Surgical Procedures, Blood Pressure (BP) Determination, Microscopy and ImageJ Cell Counting. 

We have refined the wording. These things are standard in the field. We added the summary diagram to help.

Male Long Evans rats were purchased from Envigo mentioned in the section Breeding Protocol. I think it should be mentioned earlier when the rats were mentioned for the first time.

The males are introduced in this section because that is where they are used and are appropriate there.

 Line 156, blocked with 2% donkey serum in KPBS with triton x-100. Please mention the percentage of triton x-100 and how long they were blocked.

Time was added.

Please mention the amount of RNA used in RNA processing and real-time PCR.

The cocktail used for RNA processing was included now.

Line 213-216, Figure 1. CardiovascularMeasures.A)MeanarterialbloodpressureinmmHg.(B)Meanarterialbloodpressureduringtheactivephase.C)Meanarterialbloodpressureduringtheinactivephase.D)Terminalheartweightnormalizedtobodyweight.*p<0.05.Dataarepresentedasmean±SEMandanalyzedbytwowayANOVA.Maineffectsofdietsandmaternalsurgeryarereported. Please....

I am not sure what is being asked here. Its unclear.

Please clearly show the significance (i.e., *) on all figures, clearly show compared to which group the data is significant.

We have edited the figures to show significance more clearly.

Immunohistochemistry figures are not clear enough, please provide big clear images, label them for the staining beside the images, mention or show the changes in the figure by using arrows or some other things.

We have made these suggestions.

Reviewer 2 Report

Comments to the Authors of manuscript number: brainsci-2477139 entitled “Reproductive, cardiovascular and appetitive indices in female offspring of maternal vertical sleeve gastrectomy”.

Reading the introduction someone expected a very good study, however the part of methods and further results leave the reader disappointed. Why? The introduction clearly present the problem concerning the obese females, while the rats did not differ in the weight. The question rises. Why the surgery was performed? In this study not the dies is tested but the effect of VSG performed in obese dams on the future their offspring. If this goal was not achieved these data are not useful.

1. L 15 rephrase this part, avoid to use “we” or “our lab”

2. L 24 abbr.

3. L 44 space

4. L 62  punctuation

5. L 65 – abbr. was introduced above thus it should be used. Uniform the text, please

6. L 78- this sentence seems to be not finished

7. L 79 time

8. L75-83 there should be presented hypothesis and what is a novelty or why this study was undertaken but not the summary

9.   L 88 – the number of the permission is needed

10. L 91 – if this study involves obese rats, the information about the body weight is confusing, the age should be added and the body weight of each dam`s group presented

11. L 101 – has it finished?

12. L 106 – is it all?

13. L 108 – 110 to remove it…what?

14. L 118 – what time after surgery this stabilization was noted?

15. L 141 – abbr. was used in the abstract. The text should be uniform

16. L 138-143 – please explain why this procedure was performed and how many times

17. L 145-150- were rats alive during this procedure?

18. L 154 – “our”

19. L 155, 156 – abbr.

20. L 162 – DAB is used for visualization, but why hydrogen peroxidase is used at this moment?

21. L 172- 176 – if the semi-quantive analysis was not performed, why the 8-bit greyscale image was used. The cells were counted only

22. L 178 – what does “terminal” mean in this case?

23. L 182- what method?

24. part 192 what is a unit in this comparison? What was n? sex of offspring?

25. L 200 – these animals subjected to surgery were not obese. It is a methodologic mistake.

Author Response

  1. L 15 rephrase this part, avoid to use “we” or “our lab” DONE.
  2. L 24 abbr. DONE
  3. L 44 space DONE
  4. L 62punctuation DONE
  5. L 65 – abbr. was introduced above thus it should be used. Uniform the text, please DONE
  6. L 78- this sentence seems to be not finished DONE
  7. L 79 time DONE
  8. L75-83 there should be presented hypothesis and what is a novelty or why this study was undertaken but not the summary DONE
  9.  L 88 – the number of the permission is needed  DONE
  10. L 91 – if this study involves obese rats, the information about the body weight is confusing, the age should be added and the body weight of each dam`s group presented, The surgery was done on HFD fed females.  IF they are too obese, they will never get pregnant later.  
  11. L 101 – has it finished? DONE
  12. L 106 – is it all? DONE
  13. L 108 – 110 to remove it…what? DONE
  14. L 118 – what time after surgery this stabilization was noted? DONE
  15. L 141 – abbr. was used in the abstract. The text should be uniform
  16. L 138-143 – please explain why this procedure was performed and how many times. DONE
  17. L 145-150- were rats alive during this procedure? Yes.
  18. L 154 – “our” Removed this.
  19. L 155, 156 – abbr. DONE
  20. L 162 – DAB is used for visualization, but why hydrogen peroxidase is used at this moment? Peroxide activates the DAB. 
  21. L 172- 176 – if the semi-quantive analysis was not performed, why the 8-bit greyscale image was used. The cells were counted only. WE edited this section.
  22. L 178 – what does “terminal” mean in this case? Changed to trunk blood
  23. L 182- what method?
  24. part 192 what is a unit in this comparison? What was n? sex of offspring? This was edited.
  25. L 200 – these animals subjected to surgery were not obese. It is a methodologic mistake. The mothers are not described here. These are offspring born after the moms have their surgery

Reviewer 3 Report

The manuscript is written very nicely but there are some lacuna which need to be corrected.

1. Some punctuation and grammatical errors are there which need to be corrected.

2. Why the male and female rats of different body weight has been used? Is there any specific reason for that?

3. In line 89, Institutional Animal Care and Use Committee approval number should be mentioned.

4. In line 116, ml is written as MLS. is it correct.

5. Figure 2 and 3 should be of high resolution.

The manuscript is written very nicely but there are some lacuna which need to be corrected.

1. Some punctuation and grammatical errors are there which need to be corrected.

2. Why the male and female rats of different body weight has been used? Is there any specific reason for that?

3. In line 89, Institutional Animal Care and Use Committee approval number should be mentioned.

4. In line 116, ml is written as MLS. is it correct.

5. Figure 2 and 3 should be of high resolution.

Author Response

The manuscript is written very nicely but there are some lacuna which need to be corrected.

Thank you

1. Some punctuation and grammatical errors are there which need to be corrected.

2. Why the male and female rats of different body weight has been used? Is there any specific reason for that? Males are only used for breeding purposes.

3. In line 89, Institutional Animal Care and Use Committee approval number should be mentioned. WE added this

4. In line 116, ml is written as MLS. is it correct. Fixed this

5. Figure 2 and 3 should be of high resolution. We improved this

Round 2

Reviewer 1 Report

Immunohistochemistry: serum in KPBS with 4% Triton x-100 in KPBS. 4% Triton or 0.4%? Please check and confirm.

Line 213-216, Figure 1. CardiovascularMeasures.A)MeanarterialbloodpressureinmmHg.(B)Meanarterialbloodpressureduringtheactivephase.C)Meanarterialbloodpressureduringtheinactivephase.D)Terminalheartweightnormalizedtobodyweight.*p<0.05.Dataarepresentedasmean±SEMandanalyzedbytwowayANOVA.Maineffectsofdietsandmaternalsurgeryarereported. There were no space between the words. But in the revised version, this problem was fixed.

Please check if there are any corrections required.

Author Response

Thank you for the correction. It is 0.4% Triton X 100

I am not sure why it looks like there are no spaces in the cardiovascular legend. It's not how it appears on my end.

Thank you Kindly.

Reviewer 2 Report

The Authors answered almost all comments, but the study design should be rephrased and it should be presented in such manner that allow to understand that dams were subjected to the surgery, then were pregnant and their offspring were studied.

Author Response

Dear Reviewer. I have added the pink highlights to demonstrate how hard we have tried to make it clear what we are reporting. I really appreciate the desire for clarity but I am not sure where and how to make it more clear.